# The Emerging Role of Circulating T Follicular Helper Cells in Dengue Virus Immunity: Balancing Protection and Pathogenesis

**DOI:** 10.3390/v17050652

**Published:** 2025-04-30

**Authors:** Paola N. Flores-Pérez, José A. Collazo-Llera, Fabiola A. Rodríguez-Alvarado, Vanessa Rivera-Amill

**Affiliations:** 1School of Medicine, Ponce Health Sciences University, Ponce 00716, Puerto Rico; pflores21@stu.psm.edu (P.N.F.-P.); jcollazo22@stu.psm.edu (J.A.C.-L.); 2Ponce Research Institute, Ponce 00716, Puerto Rico; farodriguez@psm.edu

**Keywords:** dengue virus, flavivirus, T follicular helper cells, T cells

## Abstract

Flaviviruses are a group of viruses transmitted mainly by mosquitoes and ticks, causing severe diseases in humans. Examples include dengue, Zika, West Nile virus, and yellow fever. They primarily affect individuals in tropical and subtropical regions, causing public health problems such as epidemic outbreaks and significant economic burdens due to hospitalizations and treatments. They share antigens, leading to cross-reactivity where antibodies generated against one flavivirus can react with others, complicating the accurate diagnosis of individual infections and making the development of treatments or vaccines more challenging. The role of T cells in the immune response to flaviviruses is a complex topic debated by scientists. On one hand, T cells help control infection by eliminating infected cells and protecting against disease. However, there is evidence that an excessive or dysregulated T cell response can cause tissue damage and worsen the disease, as seen in severe dengue cases. This duality underscores the complexity of the immune response to flavivirus infections, posing a significant challenge for researchers. Gaining a deeper understanding of the immune response at the cellular level, particularly the role of T follicular helper cells, can reveal new avenues of investigation that could lead to novel strategies for disease management. This review explores the dynamics of T cell responses, focusing on circulatory T follicular helper cells (cT_FH_), to enhance our understanding of flavivirus immunity and inform future interventions.

## 1. Dengue Virus Overview

Flaviviruses originated in 1984 when differences in viral structure, genome, and biology were discovered among togaviruses, and similarities in vectors were present. The genus *Flavivirus* comprises primarily dengue (DENV), yellow fever (YFV), West Nile (WN), Zika (ZIKV), hepatitis C (HCV), and Japanese encephalitis (JE) viruses, among others. These flaviviruses are classified as arboviruses due to their shared transmission by arthropod vectors [1]. The structural viral proteins encoded by this RNA genome include a membrane (M), envelope (E), and seven other non-structural (NS) proteins. The viral particle is surrounded by an outer layer of envelope (E) glycoproteins arranged as dimers, which mediate the initial attachment of the virus to the host cell. Each E protein consists of three structural domains that undergo pH-dependent conformational changes to expose a fusion loop, enabling the fusion of viral and host membranes and facilitating the release of viral RNA into the host cell. The M proteins, which help form the outer membrane of the viral particle along with E proteins, are derived from precursor membrane (prM) proteins. The presence of both mature M proteins and residual prM proteins on the viral surface contributes to the heterogeneity of viral particles and indicates viral maturity and infectivity. This variation can influence the degree of infectivity and the immune system’s ability to recognize the virus. Among the non-structural proteins, NS1 is secreted and contributes to immune evasion and vascular leakage. NS2A plays a role in RNA replication and interferon antagonism, while NS2B serves as a cofactor for the NS3 serine protease. NS3 has multiple enzymatic activities, including nucleoside triphosphatase (NTPase), RNA helicase, and protease functions. NS4A supports the formation of replication vesicles, and NS4B suppresses interferon-beta (IFN-β) and interferon-gamma (IFN-γ) signaling, thereby weakening the host’s antiviral response. NS5, the largest and most conserved protein, contains RNA-dependent RNA polymerase (RdRp) and methyltransferase domains, both essential for viral RNA synthesis and immune evasion. The prM, E, and NS1 proteins are key targets for viral neutralization and pathogenesis studies [2,3,4].

Dengue is the most common mosquito-borne viral disease caused by four different but closely related serotypes (DENV1-4), which share 65% homology but typically manifest with the same clinical presentation. Recently, DENV5 was discovered in Malaysia, but now it is rare [5,6]. The pathophysiology of the disease manifests through the attachment of the envelope protein to the host receptors that recognize the envelope protein epitopes, as well as through the recognition of anti-DENV antibodies through the FC receptor that binds the antibody isotype, leading to uptake of the virus, and eventual release of the positive-sense RNA, leading to the direct utilization and translation of viral components for maturation and release by the use of host translational machinery and the use of the host protein processing apparatus [7].

The different serotypes share the same clinical manifestations. However, there are differences between serotypes in disease severity, epidemic potential, transmissivity, and dynamics related to the mosquito vector and immune status of the patient. There is a consensus that DENV-2 may lead to higher rates of dengue hemorrhagic fever [8,9]. Some studies detail how DENV-2 of Asian origin replicates to higher titers in hosts at a higher rate than American DENV-2 strains. DENV-4 introduced in Puerto Rico in epidemics in 1982, 1986, and 1998 depicted clade changes in the circulating virus that contributed to increased transmission efficiency, where similar behavior has been seen in DENV-2 in the South Pacific and DENV-3 in Sri Lanka, with amino acid changes expressed in NS proteins. Based on observations, DENV serotypes may not be inherently virulent. However, they may be conditionally virulent based on repeat exposure, augmented by mutants or mutations with differing degrees of antibody-dependent enhancement (ADE) interactions and, thus, exhibiting different infectivity rates [8,9]. The phenomenon of antigenic sin also occurs with the dengue virus due to the cross-reactivity between serotypes. This can result in more severe disease upon subsequent exposure to a different serotype or even another flavivirus. This effect is largely associated with the envelope (E) protein and its three structural domains. Antigenic sin arises when the immune system, after initial exposure to one dengue serotype, mounts a memory response that is preferentially directed against that first serotype. Upon later infection with a different serotype, the immune system relies on this existing memory rather than generating a new, serotype-specific response. This leads to suboptimal neutralization and immune protection, contributing to enhanced disease severity—an effect similar to that seen with the influenza virus [4]. In this sense, viral immunity is achieved by forming antibodies to the antigen in the host’s humoral system. In the case of Flaviviridae, this may be achieved by the production of efficacious neutralizing antibodies (nAbs) through vaccine-mediated immunity, as is the case with YFV and the YFV vaccine (YF-17D), with attenuated (weakened) organisms introduced via injection for a long-lived response [10,11]. A major challenge in dengue vaccine development is generating nAbs that provide protection against all four serotypes without triggering antibody-dependent enhancement (ADE). This requires activating immune cells capable of recognizing a broad range of viral particles.

## 2. Dengue Clinical Manifestations

The typical clinical presentation of dengue fever presents as a patient who has an acute onset of fever, headache (or retroorbital pain), body aches, and rash spreading from the trunk, typically in patients who live in endemic areas or have traveled to endemic areas two weeks before the onset of symptoms. If a patient presents with this clinical picture, dengue is a nationally notifiable disease that should be reported to the local health department [12].

According to the updated WHO classification of 2009, dengue fever illness is categorized as 1-dengue without warning signs (previously acute dengue fever, also known as D-W), 2-dengue with warning signs (previously dengue hemorrhagic fever or DHF, now known as D+W), and 3-severe dengue (previously dengue shock syndrome or DSS, now known as SDF) [13]. D-W is present in patients who have traveled near or reside within a dengue-endemic area and present with a fever that lasts anywhere from 2–7 days in duration and has experienced one of the following: nausea, vomiting, rash, aches, or pains, a positive tourniquet test or leukopenia which require case and outpatient management on supportive treatment such as IV fluid repletion to manage volume depletion and acetaminophen as needed to manage fever [14]. D+W may present with the previous criteria around the time the fever and the body temperature decrease, and the patient also exhibits one of the following: severe abdominal pain or tenderness, vomiting, edema, orthostatic hypotension, mucosal bleeding, lethargy, restlessness, hepatomegaly, or a concurrent increase in hematocrit with a rapid decrease in platelets. At this point, hospital and observational admission are recommended, and blood transfusions may be given based on the severity of the bleeding observed [12]. SDF meets the criteria for dengue with warning signs and may also have one of the following: severe plasma leakage leading to shock or edema with respiratory distress, severe bleeding from the gastrointestinal tract or vaginal requiring IV fluid resuscitation or emergent blood transfusion, or severe organ impairment as evidenced by elevated transaminases ≥ 1000 IU/L, impaired consciousness, or heart impairment, in which case ICU admission is required [15].

The disease is thought to develop in three distinct phases, composed of a febrile phase, a critical phase, and a recovery phase. In both classification schemes, acute dengue fever manifests with febrile and recovery phases. In contrast, the moderate hemorrhagic presentation with warning signs and severe shock syndrome variants exhibits a febrile phase, a critical phase, and a recovery phase [16]. The symptoms mentioned above for the acute presentation in both classification schemes can describe the febrile phase, in which a patient rarely experiences a remission in fever and may recur for 1–2 days (bi-phasic fever) [14,17,18]. A physical exam with a tourniquet test screening for petechiae and a CBC with liver enzymes will be able to screen for the criteria mentioned above and for intervention if the patient needs correction [19,20]. It is recommended that the progression to plasma leakage in a patient, especially on days 3 and 7 of illness, be monitored using ultrasonography of the chest and abdomen and radiography of the chest [21,22]. The critical phase may occur more than 18 months after the initial infection of dengue after resolution or in children under one year of age with low levels of maternal antibodies, and the child may experience the infection as a primary infection and progress directly to the critical phase [16]. This may also be seen in patients with significant immunosuppressive comorbidities. The critical phase itself lasts from 24–48 h. It should involve carefully monitoring blood pressure with attention to careful resuscitation, as the patient may experience rapid-onset hypotension that may lead to shock. The recovery phase is marked by a resolution of plasma leakage and bleeding, resorption of excess fluids, and vital sign stabilization. However, an additional rash lasting at most five days may manifest after the resolution of the fever, and the patient may experience fatigue for days or weeks after recovery. The recovery phase lasts approximately 2–3 days [12].

## 3. Contribution of CD8+ T Cells to the Immune Response Against Dengue Virus

In recent years, significant advancements have been made in understanding the role of T cells in the immune response to dengue virus (DENV). Both CD8+ cytotoxic T cells and CD4+ helper T cells are indispensable in controlling DENV infection. The immune response is characterized by robust antiviral T cell activation, with CD8+ T cells targeting non-structural proteins like NS3, NS4b, and NS5, while CD4+ T cells focus on structural proteins such as the capsid, envelope, and secreted non-structural protein NS1 [23]. This complementary targeting ensures a coordinated immune defense: CD8+ T cells act as effectors that eliminate infected cells, and CD4+ T cells amplify the antiviral response by activating other immune cells and enhancing antibody production. The activation of CD8+ T cells is initiated when viral peptides, typically 9–10 amino acids in length, are processed and presented by MHC class I molecules on infected cells. CD8+ T cells recognize these peptide-MHC complexes via their T cell receptors (TCR), triggering the release of cytotoxic granules containing perforin and granzymes. These granules induce apoptosis in infected cells, thereby limiting viral replication and contributing to immune protection [23,24]. Beyond cytotoxicity, CD8+ T cells secrete antiviral cytokines such as IFN-γ and tumor necrosis factor-alpha (TNF-α), which activate macrophages and dendritic cells and establish a systemic antiviral state that curtails viral spread [25,26].

Memory CD8+ T cells generated during primary infections play a pivotal role in protecting against subsequent infections by the same DENV serotype. These memory cells enable a rapid and robust immune response that reduces disease severity and prevents progression to severe disease manifestations [25,26]. An intriguing aspect of CD8+ T cell responses in dengue is their skin-homing phenotype. During acute infection, CD8+ T cells express cutaneous lymphocyte antigen (CLA), which indicates that the skin is a critical site for immune surveillance and may be linked to dengue-associated dermatological symptoms [27]. CD4+ helper T cells complement this response by recognizing longer viral peptides (12–15 amino acids) presented by MHC class II molecules on antigen-presenting cells (APCs) such as dendritic cells and macrophages. These helper T cells enhance immune defense by promoting the activation of B cells, aiding in antibody production, and supporting CD8+ T cell responses [28]. CD4+ T cells are further categorized into subsets—Th1, Th2, Th17, and Treg—each contributing uniquely to the immune response. Th1 cells, for instance, secrete IFN-γ to promote cell-mediated immunity, while Th2 cells produce interleukin (IL)-4 and IL-5 to support humoral responses [23,29]. Interestingly, CD4+ T cells also display direct cytotoxicity akin to CD8+ T cells, killing infected cells and producing inflammatory cytokines to control the infection [25].

Recent studies have identified highly polarized CX3CR1+ cytotoxic CD4+ T cells in dengue patients, which exhibit potent antiviral activity and could serve as targets for novel vaccine strategies [30,31]. However, as with CD8+ T cells, CD4+ T cell responses must be tightly regulated to prevent immunopathogenesis. Dysregulated T cell responses during secondary infections may exacerbate disease severity, particularly in the context of ADE, where preexisting non-neutralizing antibodies facilitate viral entry into host cells [32,33]. Balancing the protective and potentially pathogenic roles of T cells is essential for vaccine development. The dual roles of CD8+ and CD4+ T cells in protection and immunopathogenesis underscore the complexity of dengue immunity. Effective vaccines must elicit robust and balanced T cell responses that provide cross-serotype protection while minimizing the risk of severe disease. Current research focuses on optimizing epitope selection and inducing memory T cell responses to achieve these goals [26,31].

## 4. CD4+ T Cells and Their Contribution to Dengue Virus Immunity

CD4+ T cells play a central role in orchestrating the immune response against dengue virus (DENV) infection. They recognize antigens presented by major histocompatibility complex (MHC) class II molecules on the surface of antigen-presenting cells (APCs) such as dendritic cells, macrophages, and B cells. This recognition not only activates CD4+ T cells but also enables them to facilitate the activation of other immune cells and produce cytokines that modulate the immune response, ensuring efficient immune functioning [23,28]. CD4+ T cells are functionally divided into subsets, including Th1, Th2, Th17, and Treg cells. Each subset has a distinct role in immune regulation. Th1 cells secrete IFN-γ, which activates macrophages and promotes cytotoxic T cell responses, forming the backbone of cell-mediated immunity. Th2 cells release IL-4 and IL-5, aiding B cells in producing antibodies and enhancing humoral immunity. Th17 cells specialize in combating extracellular bacteria and fungi, while Treg cells maintain immune tolerance, preventing autoimmune damage [29]. These subsets collectively ensure a balanced immune response.

In dengue, CD4+ T cells primarily recognize peptides of 12–15 amino acids presented by APCs on MHC class II molecules. This interaction is critical for eliciting robust humoral responses, including the production of high-affinity nAbs against DENV. Furthermore, CD4+ T cells contribute to establishing immunological memory by assisting B cells in producing long-lived plasma cells and supporting CD8+ T cells in generating memory T cells. These mechanisms enhance the immune system’s ability to respond more efficiently to subsequent DENV exposures [27,31]. In addition to their helper roles, CD4+ T cells produce pro-inflammatory cytokines such as IFN-γ and TNF-α. These cytokines reinforce the antiviral state in infected cells and amplify immune responses by recruiting and activating other immune cells. Importantly, CD4+ T cells also exhibit direct cytotoxic activity, a capability previously thought to be restricted to CD8+ T cells. By killing virus-infected cells, CD4+ T cells provide an additional immune defense against DENV [28,30].

Recent studies have uncovered the significance of highly polarized CX3CR1+ cytotoxic CD4+ T cells in dengue. These cells demonstrate potent antiviral activity by eliminating infected cells and secreting inflammatory cytokines to control the infection. A study by Weiskopf et al. revealed that these polarized T cells are critical for protective immunity and represent promising targets for vaccine development [30,31]. Inducing similar T cell responses could enhance vaccine efficacy by leveraging both cytotoxic and helper functions of CD4+ T cells. Despite their protective roles, CD4+ T cells can contribute to immunopathogenesis during secondary DENV infections. Imbalanced responses, particularly in the presence of ADE, can exacerbate disease severity, leading to complications such as DHF or DSS [32,33]. Balancing protective immunity with the prevention of adverse immune responses is essential for managing dengue and optimizing vaccine strategies. The dual role of CD4+ T cells in protection and immunopathogenesis underscores the challenges of dengue vaccine development. Vaccines must elicit protective immune responses without triggering severe disease outcomes. Future strategies, such as epitope-based vaccines and modulation of T cell subsets, aim to achieve this balance and improve the prospects of long-term immunity against DENV [27,31].

## 5. The Role of Circulatory T Follicular Helper Cells

Interactions between B and T cells are required to effectively generate protective nAbs and immunological memory [34]. This interaction primarily occurs in the germinal centers of secondary lymphoid organs [35]. During the germinal center reaction, a subset of cognate CD4+ T cells, the T follicular helper (T_FH_) cells, help activate and differentiate B cells and help in antibody production during infection (Figure 1) [36]. As a result, the chosen high-affinity B cells could differentiate into plasmablasts, plasma cells, or memory B cells. In secondary lymphoid tissues, T_FH_ cells enhance CXCR5 expression while reducing CCR7 expression, allowing them to migrate to the follicles and aid B cells [37]. The secretion of IL-21 by T_FH_ cells stimulates the differentiation and class-switching of B cells [38]. Studying these interactions in vivo in humans with a realistic microenvironment is challenging with existing experimental systems and is invasive for the study participants. However, circulating CD4+ T cells expressing CXCR5, named circulating T_FH_ (cT_FH_) cells, have been suggested as a peripheral counterpart of T_FH_ cells with similar functions and are easier to study in the context of immune responses to pathogens (Figure 1) [39].

Circulatory T follicular helper cells express the surface protein PD-1, enabling them to receive signals from B cells via the PD-1 ligands (PD-L), which regulate the activity of cT_FH_ cells [40,41]. The cT_FH_ cells express the inducible costimulator (ICOS), a co-stimulatory molecule critical in many aspects, including development, follicle migration, and functions [41]. These cT_FH_ cells also help B cells during the germinal center reaction by re-entering the lymph nodes in the case of re-infection [42]. This interaction between T follicular helper cells and B cells is essential for generating high-affinity antibodies and developing effective immune responses [43]. The cT_FH_ cells are divided into different subsets. The combinations of these markers ultimately define them: CXCR5, CXCR3, CCR6, PD-1, and ICOS [11,39,44,45]. The combination of these markers defines the cT_FH_ subsets, where mainly the combination of the markers is CXCR5+ CXCR3+ CCR6- cT_FH1_ cells, CXCR5+ CXCR3- CCR6- cT_FH2_ cells, and CXCR5+ CXCR3- CCR6+ cT_FH17_ cells [44,46]. These subsets exhibit distinct cytokine profiles and play different roles in immune responses. The ICOS and PD-1 expressions on these cells present different capacity levels at which these cells can help B cells [47]. A growing body of evidence has indicated that CXCR3−cT_FH_ cells in human blood, comprising cT_FH2_ and cT_FH17_ subsets, serve as crucial functional equivalents for GC T_FH_ cells because these cells effectively stimulate naïve B cells to generate antibodies in vitro; hence, they are known as “efficient helper cells”. On the other hand, CXCR3+ cT_FH_ cells, comprising the cT_FH1_, do not possess this capability, which is why they are known as “non-efficient helper cells”. However, this “efficiency” in helping B cells seems to vary depending on the pathogen causing the infection.

The CXCR5+, CXCR3+, CCR6-, PD-1+, and ICOS+ cT_FH1_ cells are associated with promoting B cell responses by helping the memory B cells (MBCs), but not the naïve B cells, in becoming antibody-producing cells and producing IgG antibodies [48,49]. The cT_FH1_ produces the cytokines IFN-γ and IL-21, which help activate B cells and promote class switching. What promotes cT_FH1_ differentiation is IL-12, IFN gamma by activating the transcription factor (TF) T-bet, IL-2 deprivation, TCR signaling, ICOS signal, and CD28 signal [50,51]. The transcription factors responsible for promoting cT_FH1_ activation are T-bet and Bcl6 [52,53]. The CXCR5+, CXCR3-, CCR6-, PD-1+, and ICOS+ cT_FH2_ cells promote B cell responses characterized by the production of IgE and IgG antibodies, and they produce IL-4, IL-5, and IL-13 [51]. The cT_FH2_ differentiation occurs by IL-4 by activating TF GATA3, ICOS and CD28, IL-2 deprivation, IL-6, and IL-21. The transcription factors that promote cT_FH2_ activation are GATA3 and Bcl6 [47,54]. The CXCR5+, CXCR3-, CCR6+, PD-1+, and ICOS+ cT_FH17_ cells are associated with promoting B cell responses characterized by the production of IgA antibodies [41]. They produce the cytokines IL-17 and IL-21, which help activate B cells and support class switching to IgA and IgG. This cT_FH17_ differentiation is mediated through IL-6 and TGF-β (together activate the TF RORγt), IL-21, ICOS, CD28, and IL-23 (produced by APCs, which promote TF RORγt) [54,55]. RORγt is a master regulator of cT_FH17_ differentiation and promotes IL-17 and IL-22 production [51]. In summary, the cT_FH_ cells are a specialized subset of CD4+ T cells that assist B cells in producing high-affinity antibodies, crucial for effective immunity. Originating from naïve T cells and influenced by factors like IL-21 and Bcl-6, they interact with B cells in germinal centers and secrete cytokines such as IL-21, which is important when helping B-cells. Their study is essential for understanding autoimmune diseases, improving cancer immunotherapies, and managing infectious diseases. Research on cT_FH_ cells focuses on their molecular mechanisms, clinical applications, and potential as biomarkers, aiming to harness their role in immune regulation and long-term immunity. However, more research is needed to fully understand their functions and therapeutic potential.

## 6. Characterizing Circulating T Follicular Helper Cells in Flavivirus-Induced Immune Responses

Studies regarding cT_FH_ cells suggest that these cells promote antibody responses across viral infections, including influenza, HIV-1, and SARS-CoV-2 [42,56,57]. These subsets of cT_FH_ cells, characterized by markers like ICOS, PD-1, and CXCR5, interact with B cells to activate, proliferate, and differentiate them into antibody-producing cells. The frequency and functionality of cT_FH_ cells correlate with the quality and durability of antibody responses against these viruses, underscoring their importance in immunity. Additionally, these studies emphasized the significance of nAbs in the immune response against viruses, with specialized subsets of T_FH_ cells, such as ICOS+PD-1+CXCR3+ cT_FH1_ cells, contributing to their generation by promoting B cell activation and differentiation [42,46,56,57]. The nAbs antibodies specifically bind to flaviviruses and block their ability to infect cells, being crucial for protection and the main objective of vaccines [58]. Non-neutralizing antibodies, although they mark the virus for immunological recognition, do not prevent infection and may contribute to ADE, increasing the severity of the disease. Flaviviral vaccine development focuses on inducing strong nAb responses using live-attenuated, inactivated, and recombinant vaccines while addressing challenges like ADE and ensuring durable immunity. However, scientists must also concentrate on peptide-based vaccines, which offer a promising and safe approach to developing immunity against flaviviruses by using specific viral protein sequences to elicit targeted immune responses. Understanding these cell interactions and how the cT_FH_ helps in the nAbs development could be crucial for effectively developing vaccines and therapeutic interventions to combat viral diseases.

Current research on flaviviruses focuses primarily on studying circulating T follicular helper (cT_FH_) cell populations following vaccination. For example, a study by Sandberg et al. [59] investigating the immune response following YFV 17D vaccination brought significant findings across various immune components to light. Th1-polarized CXCR3+ cT_FH_ cells exhibited heightened activation on day 14, suggesting that robust germinal center activity is essential for generating high-quality antibodies. Plasmablast expansion, particularly the IgG1 subset, peaked on day 14, correlating positively with the subsequent development of nAbs on day 90. This linkage between cT_FH1_, plasmablast frequencies, and protective immunity highlights the potential of these cells as early indicators of vaccine efficacy. Moreover, YFV-E-specific IgG1 plasmablasts, detectable by day seven and peaking at day 14, emphasize the vaccine’s ability to induce antigen-specific B cell responses. This comprehensive understanding of cT_FH_ subsets, plasmablast dynamics, memory B cell responses, and their correlations with antibody production contributes valuable insights into the mechanisms underpinning protective immunity elicited by YFV 17D vaccination [59]. Studies like this one, exploring the diverse subsets of cT_FH_ cells, provide valuable insights into how individual immune responses are tailored to combat specific pathogens based on the pathogen’s characteristics and the epitopes’ specificity. By investigating the distinct phenotypic and functional attributes of cT_FH_ subsets, researchers can elucidate how these cells modulate B cell activation and antibody production in response to various flaviviruses.

An important finding from another study on yellow fever vaccination dynamics is the correlation between the frequencies of cT_FH_ cell subsets and the strength of the nAb response post-vaccination by Huber et al. Specifically, the study revealed that the frequencies of cT_FH1_-polarized cells positively correlated with the level of nAb activity, whereas the frequencies of cT_FH17_ cells exhibited an inverse correlation. This could suggest that the composition and polarization of cT_FH_ cell subsets, particularly the cT_FH1_ subset, may serve as a predictive biomarker for the magnitude of the nAb response following yellow fever vaccination. The dominance of the cT_FH1_ subgroup, characterized by CXCR3 expression and IFN-γ secretion, was associated with higher neutralizing activity in the serum of vaccinees. At the same time, the cT_FH17_ subset, known for IL-17 production, showed a negative correlation with neutralizing antibody levels [11]. This correlation underscores the significance of cT_FH_ cell subset dynamics in shaping the humoral immune response, particularly regarding antibody production and vaccine-induced immunity.

In the context of human leukocyte antigens (HLA; the human major histocompatibility complex) molecules and HLA restriction, the study provides insights into their significant role in the function of cT_FH_ cells. By recognizing antigenic peptides presented by MHC class II molecules, cT_FH_ cells can specifically target and interact with antigen-presenting cells. The study identifies the frequencies of cT_FH1_ cells as a predictive biomarker for the level of neutralizing antibody responses, suggesting that the interaction between HLA molecules and cT_FH_ cells may influence the strength and quality of the antibody response post-YFV-17D vaccination. Through their engagement with HLA-presented viral epitopes, cT_FH1_ cells help B cells, driving their differentiation into antibody-secreting cells and promoting high-affinity antibodies to neutralize the virus [11]. Together, this can help us understand the relationship between cT_FH_ cell subsets, HLA restriction, and antibody production and can provide valuable insights for optimizing vaccination strategies. Specifically, these observations suggest that the cT_FH_1 subset effectively supports B cells in producing antibodies targeting flavivirus infections by recognizing pathogen-specific epitopes presented by HLA molecules.

Research regarding DENV, for example, by Izmirly et al. [45], investigated the role of cT_FH_ cells in the immune response pre- and post-vaccination with the TV003 vaccine, mainly focusing on their correlation with antibody production and vaccine efficacy. The study deepens into the role of circulating cT_FH_ cells in the immune response before and after vaccination, revealing several interesting insights. First, higher baseline/pre-vaccination frequencies of total cT_FH_ cells are positively correlated with neutralizing antibody titers and the breadth of the vaccine response, indicating the predictive value of cT_FH_ cells for dengue vaccine outcomes. This association extends to subjects’ PBMCs, where the frequency of total cT_FH_ cells predicts vaccine outcome and correlates with protection against all four dengue serotypes elicited by the vaccine. Additionally, a significant positive correlation exists between cT_FH_ frequencies at baseline and plasmablast frequencies post-vaccination, further emphasizing the influence of cT_FH_ cells on the immune response. The study’s implications extend to understanding natural infection with dengue virus, where prior exposure shapes cT_FH_ responses, influencing immune memory, serotype-specific immunity, and vaccine efficacy in previously infected individuals. Conducting the study in an endemic area adds clinical relevance, providing insights applicable to populations at high risk of dengue infection and enhancing the understanding of immune responses to vaccination in diverse settings. Overall, the study’s findings underscore the intricate relationship between cT_FH_ cells, B cells, and antibodies, emphasizing the importance of cT_FH_ frequencies in predicting and influencing vaccine outcomes in dengue vaccination [45].

Building on these findings, it is evident that cT_FH_ cells play a significant role in shaping the immune response after vaccination with YFV17 and the dengue vaccine TV003. However, this raises the question of how the cT_FH_ cell population behaves during natural flavivirus infection. Will the same patterns observed after vaccination also apply? In a study by Haltaufderhyde et al. [40], during acute DENV infection, there was a significant increase in the activation of cT_FH_ cells, with the cT_FH1_ subset (CXCR3+CCR6−) showing the highest activation, particularly during the critical phase of illness. Activation was indicated by elevated expression of PD-1 and co-expression of ICOS. This activation suggests that cT_FH1_ cells also have a role in the immune response during acute DENV infection, potentially influencing antibody production and overall disease outcome. Additionally, there was a positive correlation between the frequency of activated cT_FH_ cells and the frequency of plasmablasts, indicating that cT_FH_ activation may contribute to the induction of plasmablasts during acute DENV infection. Since cT_FH_ cells have a role in promoting antibody production by providing help to B cells, their activation during acute DENV infection suggests their involvement in the generation of DENV-specific antibodies. Understanding the activation of cT_FH_ cells and their role in antibody production is essential for elucidating the immune response to DENV infection. Moreover, further research on cT_FH_ cells could better understand this interaction and how these cT_FH_ subsets are involved in plasmablast activation and nAb production during a DENV infection, and for how long these nAbs last [40].

A study by Wijesinghe et al. [60] sheds light on the complex dynamics between cT_FH_ cells, plasmablasts, and antibodies during acute dengue infection, aiming to discern their impact on disease severity and clinical outcomes. Several key findings emerged. Notably, **cT_FH_ cell expansion** was more pronounced in cases of severe dengue and in secondary infections, indicating an activated immune profile. This activation was marked by elevated PD-1 expression, a known indicator of T cell activation. Further phenotypic analysis showed a significant increase in cT_FH_ cells co-expressing PD-1 and ICOS, suggesting enhanced capacity for B cell help and a potential role in modulating the humoral immune response during more severe manifestations of the disease. Functionally, IL-21-producing cT_FH_ cells surged in acute dengue, correlating strongly with plasmablasts expansion and thus driving antibody production. Plasmablast expansion, notably elevated in DHF, correlated positively with cT_FH_ cell frequency, particularly the IL-21 producers. Interestingly, cT_FH_ cells and plasmablasts frequencies inversely correlated with viral loads, indicating a role in viral control, while plasmablasts expansion correlated with DENV-specific IgG titers, especially in DHF. Nonetheless, Neut50 titers significantly correlated with cT_FH_ cell and plasmablast frequencies, notably in DHF, exhibiting a marked increase throughout convalescence, particularly in DENV2 infections. These findings could denote that the cT_FH_ cells have a role in helping B cells drive antibody responses in acute dengue infection, particularly emphasizing the significance of IL-21-producing cT_FH_ cells in generating robust nAbs. This study discusses the intricate interplay between cellular and humoral immune responses in dengue pathogenesis [60].

## 7. T Cell Function in Dengue Immune Response: Implications and Challenges for Vaccine Development

Developing a vaccine for the dengue virus (DENV) has proven exceptionally challenging due to the co-circulation of four distinct serotypes (DENV-1 to DENV-4), which share approximately 70% sequence homology. In endemic regions, individuals are frequently exposed to multiple serotypes over their lifetime. While infection with one serotype confers lifelong immunity to that specific serotype, it simultaneously increases the risk of severe dengue upon subsequent infection with a different serotype. This increased risk is attributed to ADE, wherein non-nAbs from a prior infection bind to, but fail to effectively neutralize, the secondary infecting serotype. This mechanism facilitates viral entry into host cells, amplifying infection severity [32,61]. The role of T cells in dengue immunity adds another layer of complexity. While T cells are critical in orchestrating protective immune responses and targeting infected cells, their dysregulation can contribute to immunopathogenesis. Vaccines must elicit a balanced immune response that provides broad protection without predisposing individuals to severe disease upon secondary infection. Achieving immunity against all four serotypes simultaneously is particularly difficult, as ADE and inappropriate T cell responses can exacerbate disease severity. Ensuring the long-term safety of vaccines is also crucial, as memory T cells may contribute to enhanced immunopathology in future exposures to DENV [27,33].

Researchers are exploring innovative strategies to address these challenges. One promising approach is T cell epitope mapping, which identifies conserved T cell epitopes across all DENV serotypes that elicit protective immune responses without triggering harmful immunopathology. These epitope-based vaccine designs rely on HLA-binding studies to select immunodominant epitopes that are safe and cross-reactive. Broadly protective vaccines, designed to induce durable immunity against all four serotypes, are also under investigation. Specific adjuvants and vaccine formulations are being developed to modulate immune responses and minimize ADE-associated risks. For example, some formulations favor a Th1-skewed response, which is less likely to exacerbate disease severity [28,31,62]. CD4+ T cells are particularly crucial in this context. Their activation involves antigen uptake, presentation on HLA complexes, and recognition by T cell receptors (TCRs) on naïve CD4+ T cells. This process initiates a cascade of events, including cytokine secretion and costimulatory receptor engagement, which stimulate B cells to produce nAbs. However, the complexity lies in understanding how specific CD4+ T cell subsets, such as follicular helper T cells (cTFH), contribute to either protective immunity or immunopathology. Dysregulation of these cells can drive ADE and other pathological outcomes [25,27].

Another challenge is the variability in immune responses across geographical regions. Individuals with the same HLA allele types may exhibit different magnitudes of DENV-specific CD4+ T cell activation, influenced by regional factors such as serotype prevalence and HLA polymorphism. Discrepancies between predicted and observed epitopes highlight the importance of refining HLA-restriction studies to identify universally protective epitope candidates. Comparative analyses across flaviviruses, including DENV, may also reveal conserved epitopes useful for cross-protective vaccine development [25,31]. To overcome these obstacles, researchers are integrating multiple approaches, including advanced epitope mapping, adjuvant optimization, and in-depth analysis of T cell subset dynamics. These strategies aim to enhance protective T cell responses while mitigating immunopathological risks. By improving our understanding of the intricate interplay between DENV and the immune system, it is possible to create safer and more effective vaccines. This multifaceted approach is essential to reduce the global burden of dengue and develop lasting solutions to this significant public health challenge.

## 8. Conclusions and Perspectives

Flaviviruses are a global concern, affecting millions of individuals in tropical and sub-tropical regions and causing thousands of deaths yearly. However, no completely effective treatments or vaccines currently protect entirely and safely against specific flaviviruses such as DENV. Not to mention that the vaccine currently approved for DENV by the US FDA has some limitations, including the risk of ADE in individuals without prior dengue infection, restricted efficacy across serotypes, and the need for pre-vaccination testing. In the U.S., its use is limited to children aged 9–16 who live in dengue-endemic areas and have laboratory-confirmed prior DENV infection [63,64,65]. Therefore, more research is needed to develop effective prevention methods and treatments. This review article highlights the critical role of cT_FH_ cells and antibodies in developing robust immune responses against flaviviruses, particularly DENV. Studies highlight the role of cT_FH_ cells, characterized by markers such as CXCR5, ICOS, and PD-1, in promoting antibody production and shaping the quality of immune responses to combat infections. As mentioned above, research on yellow fever 17D vaccination and DENV infection demonstrates the value of cT_FH_ cells for vaccine outcomes as a predictive method, with higher baseline frequencies correlating with improved antibody responses (Figure 2). The importance of nAbs in conferring protection against flaviviruses is emphasized, with specific cT_FH_ cell subsets, particularly those polarized toward a Th1 phenotype, playing a critical role in their generation. While significant progress has been made in understanding the functions of cT_FH_ cells, more knowledge still needs to be gained regarding certain aspects of cT_FH_ immunity, particularly in flavivirus infections. For example, it would be interesting to characterize if there is any cross-reactivity or cross-protection with the cT_FH_ cells and other flaviviruses, how long this response of the cT_FH_ lasts, the memory of the different subsets, the immunogenicity, and how the antigen presentation influences their response. While many studies have focused on correlating cT_FH_ cell fluctuations with immune responses—which is valuable—we also need functional experiments. For instance, co-culturing different cT_FH_ subsets with B cells could reveal their roles in driving B cell differentiation and the production of virus-specific and neutralizing antibodies. Additionally, assessing the cytokine profiles these cells express upon stimulation with specific peptides would help clarify the magnitude and quality of their T cell responses. Also, the role of HLA molecules in presenting viral antigens to cT_FH_ cells and in shaping the immune response warrants further investigation, as HLA polymorphisms may influence individual susceptibility to DENV infection and responses to vaccines. Closing these knowledge gaps regarding understanding the meticulous dynamics of cT_FH_ cells and their interactions with B cells will be essential to have more strategies and insights to combat flavivirus infections and improve global public health outcomes.

## Figures and Tables

**Figure 1 viruses-17-00652-f001:**
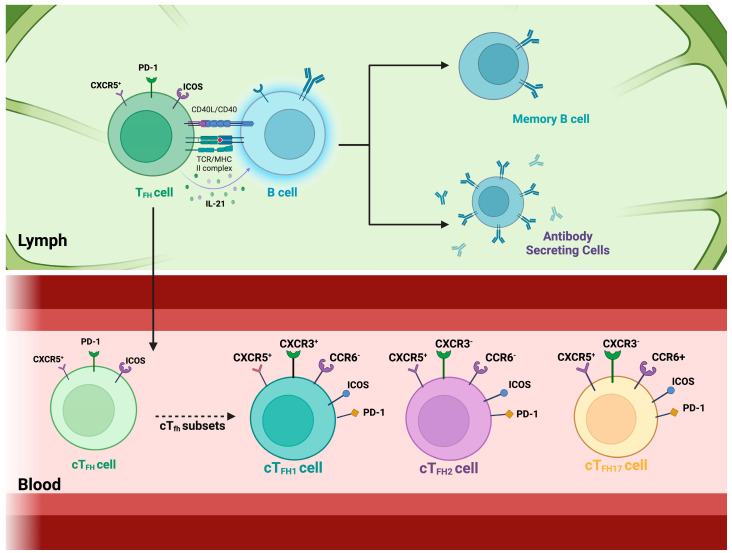
T follicular helper cells and their circulating counterparts. Top panel: T follicular helper (T_FH_) cells are a specialized subset of CD4+ T cells that support B cell activation and antibody production. These cells reside in secondary lymphoid organs, such as the tonsils, spleen, and lymph nodes, where they localize to B cell zones. By interacting closely with B cells, T_FH_ cells facilitate their activation and differentiation into antibody-secreting plasma cells and memory B cells, ensuring long-lasting immunity. This process is driven by T_FH_ cell-mediated co-stimulation through CD40 ligand (CD40L) and cytokine secretion, particularly IL-21, which promotes B cell proliferation. T_FH_ cells are defined by the transcription factor B-cell lymphoma 6 (Bcl6) and express surface markers such as CXCR5, PD-1, and ICOS. Bottom panel: Circulating T follicular helper (cT_FH_) cells represent a subset of T_FH_ cells that can be detected in the bloodstream. These cells are classified based on their expression of chemokine receptors CXCR3 and CCR6 into three major subsets: cT_FH1_ (CXCR3+CCR6−), cT_FH2_ (CXCR3−CCR6−), and cT_FH17_ (CXCR3−CCR6+). Among these, cT_FH_ cells expressing PD-1 and ICOS have been identified as the most efficient in providing B cell help, highlighting their potential role in shaping immune responses.

**Figure 2 viruses-17-00652-f002:**
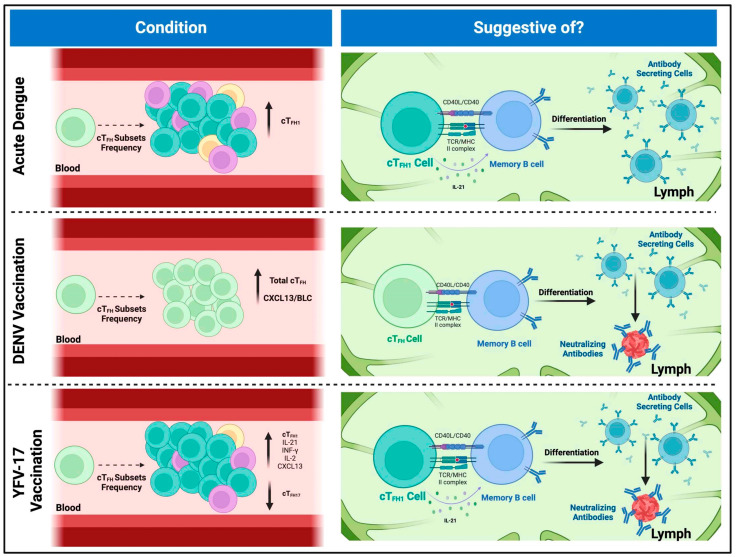
Implications of circulating T follicular helper cell frequencies in flaviviral infections and vaccine responses. Top panel: During acute dengue virus (DENV) infection, cT_FH1_ cells, characterized by the expression of PD-1, ICOS, and IL-21 production, are present at high frequencies. The frequency of these cells correlates with the presence of antibody-secreting cells. Although cT_FH1_ cells are typically classified as “non-efficient helper cells” due to their limited capacity to activate and differentiate naïve B cells, they have been shown to support the activation and differentiation of memory B cells (MBCs) into antibody-secreting plasma cells and plasmablasts. This suggests that, during acute dengue infection, cT_FH1_ cells are among the first responders, facilitating the differentiation of MBCs into antibody-producing cells upon their re-entry into secondary lymphoid organs. Middle panel: In the context of DENV TV003 vaccination, higher baseline frequencies of cT_FH_ cells are associated with more robust neutralizing antibody responses and broader vaccine-induced immunity. Also, baseline cT_FH_ levels correlate with post-vaccination frequencies of antibody-secreting cells, suggesting a direct role of cT_FH_ cells in driving the immune response. Elevated levels of CXCL13/BLC correlate with increased cT_FH_ frequencies, further complicating the relationship between cT_FH_ cells, B cell activation, and antibody production in dengue vaccination. These findings imply that cT_FH_ cells contribute to the differentiation of MBCs into neutralizing antibody-producing cells post-vaccination. Bottom panel: Research on yellow fever vaccination shows that cT_FH1_ cells peak in activation post-vaccination, coinciding with strong germinal center activity and the generation of high-quality antibodies. The secretion of cytokines such as IFN-γ, IL-2, and IL-21 by cT_FH1_ cells is linked to enhanced antibody responses. Moreover, the expansion of antibody-secreting cells like plasmablasts correlates with the development of neutralizing antibodies, indicating that cT_FH1_ cells serve as early indicators of vaccine efficacy. The composition of cT_FH_ subsets, especially the dominance of cT_FH1_ cells, predicts the magnitude of neutralizing antibody responses, with a positive correlation between cT_FH1_ prevalence and neutralizing activity.

## Data Availability

Not applicable.

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
