# Peer review of "The Emerging Role of Circulating T Follicular Helper Cells in Dengue Virus Immunity: Balancing Protection and Pathogenesis"

_viruses, 2025, doi:10.3390/v17050652_

Round 1
Reviewer 1 Report
Comments and Suggestions for Authors
This review provides a thorough examination of the emerging role of circulating T follicular helper (cTFH) cells in immune responses to flavivirus infections, with a strong emphasis on dengue virus (DENV). It integrates current knowledge on flavivirus biology, clinical disease progression, and the dual nature of T cell responses—both protective and pathogenic. The manuscript offers a detailed overview of cTFH subsets, their cytokine profiles, and their interactions with B cells in shaping antibody responses, particularly the generation of neutralizing antibodies. By highlighting key studies in both vaccination and natural infection contexts, the review underscores the importance of cTFH cells as potential biomarkers for vaccine efficacy and as targets for immunomodulation. This work contributes valuable insights to the ongoing efforts to develop safer and more effective flavivirus vaccines. The review is exceptionally informative and flows well, but it could benefit from some sentence structuring and more clarity.
Minor Suggestions:
1. Abbreviations should be noted alongside the full names where they first appear in the text. Any other subsequent mention of the specific name should use the abbreviated form. For instance, in line 254 Interleukin-21 (IL-21) appears for the first time but IL-21 has been mentioned throughout the article. Similarly, Interferon-gamma (IFN-γ ) appears for the first time in line 52. Subsequent mentions should use the abbreviated form.
2. Overall restructuring of long complex sentences is needed to improve clarity and better understanding. For example,
Line 34-36: ‘These Flaviviruses share similarities in transmission by the same vectors, owing to their classification of Arboviruses due to arthropod-borne disease primarily via Aedes spp. vectors and other mosquitos and ticks.’. This sentence can written as follows: ‘These flaviviruses are classified as arboviruses due to their shared transmission by arthropod vectors…’
Lines 93-98 can be restructured as follows: ‘A major challenge in dengue vaccine development is generating neutralizing antibodies that provide protection against all four serotypes without triggering antibody-dependent enhancement (ADE). This requires activating immune cells capable of recognizing a broad range of viral particles’.
The authors need to proofread the entire text and restructure complex sentences as needed.
Author Response
Comments 1: Abbreviations should be noted alongside the full names where they first appear in the text. Any other subsequent mention of the specific name should use the abbreviated form. For instance, in line 254 Interleukin-21 (IL-21) appears for the first time but IL-21 has been mentioned throughout the article. Similarly, Interferon-gamma (IFN-γ) appears for the first time in line 52. Subsequent mentions should use the abbreviated form.
Response 1: Thank you for pointing this out. We agree with this comment, so we have updated the manuscript accordingly. The corrections are tracked changes in the resubmitted files. All abbreviations are now introduced with their full terms upon first mention in the manuscript.
Comments 2: Overall restructuring of long complex sentences is needed to improve clarity and better understanding. For example,
Line 34-36: ‘These Flaviviruses share similarities in transmission by the same vectors, owing to their classification of Arboviruses due to arthropod-borne disease primarily via Aedes spp. vectors and other mosquitos and ticks.’ This sentence can be written as follows: ‘These flaviviruses are classified as arboviruses due to their shared transmission by arthropod vectors…’
Lines 93-98 can be restructured as follows: ‘A major challenge in dengue vaccine development is generating neutralizing antibodies that provide protection against all four serotypes without triggering antibody-dependent enhancement (ADE). This requires activating immune cells capable of recognizing a broad range of viral particles.
The authors need to proofread the entire text and restructure complex sentences as needed.
Response 2: Agree. We have, accordingly, revised the manuscript to address complex sentences. Updated text in the manuscript is in tracked changes. The text of the specific examples provided now reads:
Lines 35-36: These flaviviruses are classified as arboviruses due to their shared transmission by arthropod vectors.
Lines 188-191: A major challenge in dengue vaccine development is generating neutralizing antibodies that provide protection against all four serotypes without triggering antibody-dependent enhancement (ADE). This requires activating immune cells capable of recognizing a broad range of viral particles.
Reviewer 2 Report
Comments and Suggestions for Authors
The Flaviviruses are prevalent in tropical and sub-tropical regions and transmitted by mosquitoes and ticks. Major members of the group include Dengue (DENV), Zika (ZIKV), and Japanese encephalitis (JE) viruses. There are numerous similarities in the mode of transmission and disease manifestations caused by infection by these viruses and they share antigens, resulting in antibody cross-reactivity. These commonalities significantly complicate both the diagnosis of the responsible virus in infections and, in turn, the development of antiviral strategies. These complications are particularly impactful for DENV, as it exists as four serotypes that differ with respect to disease severity, epidemic potential and transmissibility. Indeed, DENV can result in three different levels of disease severity, including: 1) Dengue without Warning Signs (Acute Dengue Fever); 2) Dengue with Warning Signs (Dengue Hemorrhagic Fever); and 3) Severe Dengue (Dengue Shock Syndrome). Indeed, it is now well established that DENV disease severity is linked to repeated exposure, especially to different serotypes of the virus. This is because non-neutralizing antibodies can bind and mediate virus entry via antibody-dependent enhancement (ADE). Therefore, any vaccine approach must achieve the appropriate balance between stimulating a pan-DENV neutralizing antibody response, while not promoting ADE.
This review focuses on an area that has garnered significant interest recently, the role of T cells in Flavivirus immunity, particularly for DENV. The complexity of this topic is compounded by the reality that, while T cells can be powerful mediators of infection control by eliminating infected cells, they can also mediate inflammation, cause tissue damage and lead to the most severe of DENV manifestations. Following an initial overview of Flavivirus (almost exclusively DENV) structure and function, transmission, serotypical differences/cross-reactivity and the scope of disease manifestations, the authors take the reader through all aspects of the T cell response to the virus. This discussion of DENV and how T cells of different types control or enhance the course of the disease is considered exceptionally broad in scope, covering not only the roles of the expected CD4+ and CD8+ cells, but also the interactions between T and B cells, in particular the role of the former in stimulating the production of antibodies by B cells.
I found this manuscript to be extremely comprehensive and highly enlightening. While there are a few minor points that the authors may wish to address (see below), the review is regarded as a major contribution to the DENV field with its discussion of novel aspects of the T cell immune response and how they may be exploited to develop new antiviral approaches to control the virus without promoting disease severity.
Major topics of the review include the importance of neutralizing vs. non-neutralizing antibodies in disease progression, the role of circulating T follicular helper cells (cTFH) in the activation and differentiation of B cells to produce antibody, the combination of markers that distinguishes them and the potential for cross-reactivity with other flaviviruses and potential strategies that can be used to develop antiviral approaches based upon this T cell compartment. Strengths of this section of the manuscript include the: 1) attention to detail and abundant clarity; 2) discussion of the possibility that the roles of cTFH in vaccinated vs. naturally infected individuals may not be the same and must be investigated; 3) importance of understanding the significance of the frequencies of different cTFH subsets in the strength of the neutralizing antibody response; and, 4) innovative strategies (T cell epitope mapping, HLA binding studies, geographical variations in serotype prevalence, HLA polymorphism and adjuvant optimization and peptide-based vaccines) presently under investigation with the goal of creating vaccine approaches that will elicit protective T cell responses without stimulating the immunopathology found in individuals experiencing secondary infections.
There are a few minor points that, if addressed, may even further bolster an already strong manuscript:
- The review is understandably far more devoted to DENV than the other Flaviviruses, which should somehow be reflected in the title of the manuscript;
- The statement is made (lines 513-4) that the DENV vaccine currently approved by the US FDA has some limitations. Please provide more details about these limitations.
- The sentence in lines 38-42 makes it sound like the external layer of E proteins carry the genetic information. This sentence is poorly worded and should be improved.
- Lines 22-24 in the abstract: an understanding of the immune response at the cellular level, in and of itself, does really constitute an alternative approach of research, though it has the potential to identify new avenues of investigation that could lead to novel approaches.
Author Response
Comments 1: The review is understandably far more devoted to DENV than the other Flaviviruses, which should somehow be reflected in the title of the manuscript.
Response 1: Thank you for pointing this out. We agree with this comment. Therefore, we have updated the manuscript title to reflect the content better. The title of the review article now reads: The Emerging Role of Circulating T Follicular Helper Cells in Dengue Virus Immunity: Balancing Protection and Pathogenesis
Comments 2: The statement is made (lines 513-4) that the DENV vaccine currently approved by the US FDA has some limitations. Please provide more details about these limitations.
Response 2: Accordingly, we revised the statement to provide more details about these limitations.
Lines 660-663: Not to mention that the vaccine currently approved for DENV by the US FDA has some limitations, including the risk of antibody-dependent enhancement (ADE) in individuals without prior dengue infection, restricted efficacy across serotypes, and the need for pre-vaccination testing. In the U.S., its use is limited to children aged 9-16 who live in dengue-endemic areas and have laboratory-confirmed prior DENV infection [63-65].
Comments 3: The sentence in lines 38-42 makes it sound like the external layer of E proteins carry the genetic information. This sentence is poorly worded and should be improved.
Response 3: We have, accordingly, revised the sentence for clarity. The sentence now reads:
Line numbers 37-75: The viral particle is surrounded by an outer layer of envelope (E) glycoproteins arranged as dimers, which mediate the initial attachment of the virus to the host cell. Each E protein consists of three structural domains that undergo pH-dependent conformational changes to expose a fusion loop, enabling the fusion of viral and host membranes and facilitating the release of viral RNA into the host cell. The M proteins, which help form the outer membrane of the viral particle along with E proteins, are derived from precursor membrane (prM) proteins. The presence of both mature M proteins and residual prM proteins on the viral surface contributes to the heterogeneity of viral particles and indicates viral maturity and infectivity. This variation can influence the degree of infectivity and the immune system's ability to recognize the virus.
Comments 4: Lines 22-24 in the abstract: an understanding of the immune response at the cellular level, in and of itself, does really constitute an alternative approach of research, though it has the potential to identify new avenues of investigation that could lead to novel approaches.
Response 4: We have revised the statement for clarity and now it reads:
Lines 21-25: This duality underscores the complexity of the immune response to flavivirus infections, posing a significant challenge for researchers. Gaining a deeper understanding of the immune response at the cellular level, particularly the role of T follicular helper cells, can reveal new avenues of investigation that could lead to novel strategies for disease management.